# Reinforcement Learning Based Graph-to-Sequence Model for Natural Question Generation

**Yu Chen**
Department of Computer Science
Rensselaer Polytechnic Institute
cheny39@rpi.edu

**Lingfei Wu**[*]
IBM Research
lwu@email.wm.edu

**Mohammed J. Zaki**
Department of Computer Science
Rensselaer Polytechnic Institute
zaki@cs.rpi.edu

## Abstract

Natural question generation (QG) aims to generate questions from a passage and an answer. Previous works on QG either (i) ignore the rich structure information hidden in text, (ii) solely rely on cross-entropy loss that leads to issues like exposure bias and inconsistency between train/test measurement, or (iii) fail to fully exploit the answer information. To address these limitations, in this paper, we propose a reinforcement learning (RL) based graph-to-sequence (Graph2Seq) model for QG. Our model consists of a Graph2Seq generator with a novel Bidirectional Gated Graph Neural Network based encoder to embed the passage, and a hybrid evaluator with a mixed objective combining both cross-entropy and RL losses to ensure the generation of syntactically and semantically valid text. We also introduce an effective Deep Alignment Network for incorporating the answer information into the passage at both the word and contextual levels. Our model is end-to-end trainable and achieves new state-of-the-art scores, outperforming existing methods by a significant margin on the standard SQuAD benchmark.

## 1 Introduction

Natural question generation (QG) has many useful applications such as improving the question answering task (Chen et al., 2017; 2019a) by providing more training data (Tang et al., 2017; Yuan et al., 2017), generating practice exercises and assessments for educational purposes (Heilman & Smith, 2010; Danon & Last, 2017), and helping dialog systems to kick-start and continue a conversation with human users (Mostafazadeh et al., 2016). While many existing works focus on QG from images (Fan et al., 2018; Li et al., 2018) or knowledge bases (Serban et al., 2016; Elsahar et al., 2018), in this work, we focus on QG from text.

Conventional methods (Mostow & Chen, 2009; Heilman & Smith, 2010; Heilman, 2011) for QG rely on heuristic rules or hand-crafted templates, leading to the issues of low generalizability and scalability. Recent attempts have been focused on exploiting Neural Network (NN) based approaches that do not require manually-designed rules and are end-to-end trainable. Encouraged by the huge success of neural machine translation, these approaches formulate the QG task as a sequence-to-sequence (Seq2Seq) learning problem. Specifically, attention-based Seq2Seq models (Bahdanau et al., 2014; Luong et al., 2015) and their enhanced versions with copy (Vinyals et al., 2015; Gu et al., 2016) and coverage (Tu et al., 2016) mechanisms have been widely applied and show promising results on this task (Du et al., 2017; Zhou et al., 2017; Song et al., 2018a; Kumar et al., 2018a). However, these methods typically ignore the hidden structural information associated with a word

---

[*]Corresponding author.

sequence such as the syntactic parsing tree. Failing to utilize the rich text structure information beyond the simple word sequence may limit the effectiveness of these models for QG.

It has been observed that in general, cross-entropy based sequence training has several limitations like exposure bias and inconsistency between train/test measurement (Ranzato et al., 2015; Wu et al., 2016). As a result, they do not always produce the best results on discrete evaluation metrics on sequence generation tasks such as text summarization (Paulus et al., 2017) or question generation (Song et al., 2017). To cope with these issues, some recent QG approaches (Song et al., 2017; Kumar et al., 2018b) directly optimize evaluation metrics using Reinforcement Learning (RL) (Williams, 1992). However, existing approaches usually only employ evaluation metrics like BLEU and ROUGE-L as rewards for RL training. More importantly, they fail to exploit other important metrics such as syntactic and semantic constraints for guiding high-quality text generation.

Early works on neural QG did not take into account the answer information when generating a question. Recent works have started to explore various means of utilizing the answer information. When question generation is guided by the semantics of an answer, the resulting questions become more relevant and readable. Conceptually, there are three different ways to incorporate the answer information by simply marking the answer location in the passage (Zhou et al., 2017; Zhao et al., 2018; Liu et al., 2019), or using complex passage-answer matching strategies (Song et al., 2017), or separating answers from passages when applying a Seq2Seq model (Kim et al., 2018; Sun et al., 2018). However, they neglect potential semantic relations between passage words and answer words, and thus fail to explicitly model the global interactions among them in the embedding space.

To address these aforementioned issues, in this paper, we present a novel reinforcement learning based generator-evaluator architecture that aims to: i) make full use of rich hidden structure information beyond the simple word sequence; ii) generate syntactically and semantically valid text while maintaining the consistency of train/test measurement; iii) model explicitly the global interactions of semantic relationships between passage and answer at both word-level and contextual-level.

In particular, to achieve the first goal, we explore two different means to either construct a syntax-based static graph or a semantics-aware dynamic graph from the text sequence, as well as its rich hidden structure information. Then, we design a graph-to-sequence (Graph2Seq) model based generator that encodes the graph representation of a text passage and decodes a question sequence using a Recurrent Neural Network (RNN). Our Graph2Seq model is based on a novel bidirectional gated graph neural network, which extends the gated graph neural network (Li et al., 2015) by considering both incoming and outgoing edges, and fusing them during the graph embedding learning.

To achieve the second goal, we design a hybrid evaluator which is trained by optimizing a mixed objective function that combines both cross-entropy and RL loss. We use not only discrete evaluation metrics like BLEU, but also semantic metrics like word mover's distance (Kusner et al., 2015) to encourage both syntactically and semantically valid text generation. To achieve the third goal, we propose a novel Deep Alignment Network (DAN) for effectively incorporating answer information into the passage at multiple granularity levels.

Our main contributions are as follows:

- We propose a novel RL-based Graph2Seq model for natural question generation. To the best of our knowledge, we are the first to introduce the Graph2Seq architecture for QG.
- We explore both static and dynamic ways of constructing graph from text and are the first to systematically investigate their performance impacts on a GNN encoder.
- The proposed model is end-to-end trainable, achieves new state-of-the-art scores, and outperforms existing methods by a significant margin on the standard SQuAD benchmark for QG. Our human evaluation study also corroborates that the questions generated by our model are more natural (semantically and syntactically) compared to other baselines.

## 2 AN RL-BASED GENERATOR-EVALUATOR ARCHITECTURE

In this section, we define the question generation task, and then present our RL-based Graph2Seq model for question generation. We first motivate the design, and then present the details of each component as shown in Fig. 1.

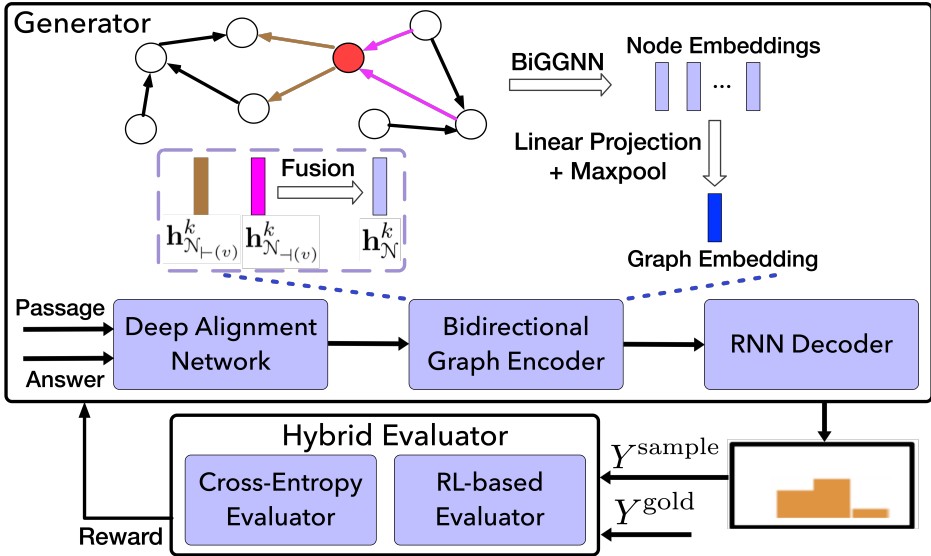

Figure 1: Overall architecture of the proposed model. Best viewed in color.

## 2.1 PROBLEM FORMULATION

The goal of question generation is to generate natural language questions based on a given form of data, such as knowledge base triples or tables (Bao et al., 2018), sentences (Du et al., 2017; Song et al., 2018a), or images (Li et al., 2018), where the generated questions need to be answerable from the input data. In this paper, we focus on QG from a given text passage, along with a target answer.

We assume that a text passage is a collection of word tokens $X^p = \{x_1^p, x_2^p, ..., x_N^p\}$, and a target answer is also a collection of word tokens $X^a = \{x_1^a, x_2^a, ..., x_L^a\}$. The task of natural question generation is to generate the best natural language question consisting of a sequence of word tokens $\hat{Y} = \{y_1, y_2, ..., y_T\}$ which maximizes the conditional likelihood $\hat{Y} = \arg\max_Y P(Y|X^p, X^a)$. Here $N$, $L$, and $T$ are the lengths of the passage, answer and question, respectively. We focus on the problem setting where we have a set of passage (and answers) and target questions pairs, to learn the mapping; existing QG approaches (Du et al., 2017; Song et al., 2018a; Zhao et al., 2018; Kim et al., 2018) make a similar assumption.

## 2.2 DEEP ALIGNMENT NETWORK

Answer information is crucial for generating relevant and high quality questions from a passage. Unlike previous methods that neglect potential semantic relations between passage and answer words, we explicitly model the global interactions among them in the embedding space. To this end, we propose a novel Deep Alignment Network (DAN) component for effectively incorporating answer information into the passage with multiple granularity levels. Specifically, we perform attention-based soft-alignment at the word-level, as well as at the contextual-level, so that multiple levels of alignments can help learn hierarchical representations.

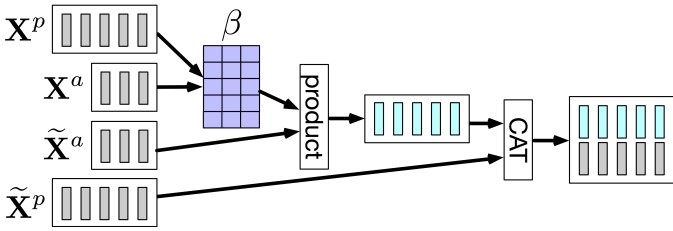

Figure 2: The attention-based soft-alignment mechanism.

Let $\mathbf{X}^p \in \mathbb{R}^{F \times N}$ and $\widetilde{\mathbf{X}}^p \in \mathbb{R}^{\widetilde{F}_p \times N}$ denote two embeddings associated with passage text. Similarly, let $\mathbf{X}^a \in \mathbb{R}^{F \times L}$ and $\widetilde{\mathbf{X}}^a \in \mathbb{R}^{\widetilde{F}_a \times L}$ denote two embeddings associated with answer text. Conceptually, as shown in Fig. 2, the soft-alignment mechanism consists of three steps: i) compute the attention score $\beta_{i,j}$ for each pair of passage word $x_i^p$ and answer word $x_j^a$: ii) multiply the attention matrix $\boldsymbol{\beta}$ with the answer embeddings $\widetilde{\mathbf{X}}^a$ to obtain the aligned answer embeddings $\mathbf{H}^p$ for the passage; iii) concatenate the resulting aligned answer embeddings $\mathbf{H}^p$ with the passage embeddings $\widetilde{\mathbf{X}}^p$ to get the final passage embeddings $\widetilde{\mathbf{H}}^p \in \mathbb{R}^{(\widetilde{F}_p + \widetilde{F}_a) \times N}$.

Formally, we define our soft-alignment function as following:

$$\widetilde{\mathbf{H}}^p = \text{Align}(\mathbf{X}^p, \mathbf{X}^a, \widetilde{\mathbf{X}}^p, \widetilde{\mathbf{X}}^a) = \text{CAT}(\widetilde{\mathbf{X}}^p; \mathbf{H}^p) = \text{CAT}(\widetilde{\mathbf{X}}^p; \widetilde{\mathbf{X}}^a \boldsymbol{\beta}^T) \tag{1}$$

where the matrix $\widetilde{\mathbf{H}}^p$ is the final passage embedding, the function CAT is a simple concatenation operation, and $\boldsymbol{\beta}$ is a $N \times L$ attention score matrix, computed by

$$\boldsymbol{\beta} \propto \exp\left(\text{ReLU}(\mathbf{W}\mathbf{X}^p)^T \text{ReLU}(\mathbf{W}\mathbf{X}^a)\right) \tag{2}$$

where $\mathbf{W} \in \mathbb{R}^{d \times F}$ is a trainable weight matrix, with $d$ being the hidden state size and ReLU is the rectified linear unit (Nair & Hinton, 2010). After introducing the general soft-alignment mechanism, we next introduce how we do soft-alignment at both word-level and contextual-level.

### 2.2.1 WORD-LEVEL ALIGNMENT

In the word-level alignment stage, we first perform a soft-alignment between the passage and the answer based only on their pretrained GloVe embeddings and compute the final passage embeddings by $\widetilde{\mathbf{H}}^p = \text{Align}(\mathbf{G}^p, \mathbf{G}^a, [\mathbf{G}^p; \mathbf{B}^p; \mathbf{L}^p], \mathbf{G}^a)$, where $\mathbf{G}^p$, $\mathbf{B}^p$, and $\mathbf{L}^p$ are the corresponding GloVe embedding (Pennington et al., 2014), BERT embedding (Devlin et al., 2018), and linguistic feature (i.e., case, NER and POS) embedding of the passage text, respectively. Then a bidirectional LSTM (Hochreiter & Schmidhuber, 1997) is applied to the final passage embeddings $\widetilde{\mathbf{H}}^p = \{\widetilde{\mathbf{h}}_i^p\}_{i=1}^N$ to obtain contextualized passage embeddings $\bar{\mathbf{H}}^p \in \mathbb{R}^{\bar{F} \times N}$.

On the other hand, for the answer text $\mathbf{X}^a$, we simply concatenate its GloVe embedding $\mathbf{G}^a$ and its BERT embedding $\mathbf{B}^a$ to obtain its word embedding matrix $\mathbf{H}^a \in \mathbb{R}^{d' \times L}$. Another BiLSTM is then applied to the concatenated answer embedding sequence to obtain the contextualized answer embeddings $\bar{\mathbf{H}}^a \in \mathbb{R}^{\bar{F} \times L}$.

### 2.2.2 CONTEXTUAL-LEVEL ALIGNMENT

In the contextual-level alignment stage, we perform another soft-alignment based on the contextualized passage and answer embeddings. Similarly, we compute the aligned answer embedding, and concatenate it with the contextualized passage embedding to obtain the final passage embedding matrix $\text{Align}([\mathbf{G}^p; \mathbf{B}^p; \bar{\mathbf{H}}^p], [\mathbf{G}^a; \mathbf{B}^a; \bar{\mathbf{H}}^a], \bar{\mathbf{H}}^p, \bar{\mathbf{H}}^a)$. Finally, we apply another BiLSTM to the above concatenated embedding to get a $\bar{F} \times N$ passage embedding matrix $\mathbf{X}$.

### 2.3 BIDIECTIONAL GRAPH-TO-SEQUENCE GENERATOR

While RNNs are good at capturing local dependencies among consecutive words in text, GNNs have been shown to better utilize the rich hidden text structure information such as syntactic parsing (Xu et al., 2018b) or semantic parsing (Song et al., 2018b), and can model the global interactions (relations) among sequence words to further improve the representations. Therefore, unlike most of the existing methods that rely on RNNs to encode the input passage, we first construct a passage graph $\mathcal{G}$ from text where each passage word is treated as a graph node, and then employ a novel Graph2Seq model to encode the passage graph (and answer), and to decode the question sequence.

### 2.3.1 PASSAGE GRAPH CONSTRUCTION

Existing GNNs assume a graph structured input and directly consume it for computing the corresponding node embeddings. However, we need to construct a graph from the text. Although there

are early attempts on constructing a graph from a sentence (Xu et al., 2018b), there is no clear answer as to the best way of representing text as a graph. We explore both static and dynamic graph construction approaches, and systematically investigate the performance differences between these two methods in the experimental section.

*Syntax-based static graph construction:* We construct a directed and unweighted passage graph based on dependency parsing. For each sentence in a passage, we first get its dependency parse tree. We then connect neighboring dependency parse trees by connecting those nodes that are at a sentence boundary and next to each other in text.

*Semantics-aware dynamic graph construction:* We dynamically build a directed and weighted graph to model semantic relationships among passage words. We make the process of building such a graph depend on not only the passage, but also on the answer. The graph construction procedure consists of three steps: i) we compute a dense adjacency matrix $\mathbf{A}$ for the passage graph by applying self-attention to the word-level passage embeddings $\widetilde{\mathbf{H}}^p$, ii) a kNN-style graph sparsification strategy (Chen et al., 2019c) is adopted to obtain a sparse adjacency matrix $\bar{\mathbf{A}}$, where we only keep the $K$ nearest neighbors (including itself) as well as the associated attention scores (i.e., the remaining attentions scores are masked off) for each node; and iii) inspired by BiLSTM over LSTM, we also compute two normalized adjacency matrices $\mathbf{A}^{\dashv}$ and $\mathbf{A}^{\vdash}$ according to their incoming and outgoing directions, by applying softmax operation on the resulting sparse adjacency matrix $\bar{\mathbf{A}}$ and its transpose, respectively.

$$\mathbf{A} = \text{ReLU}(\mathbf{U}\widetilde{\mathbf{H}}^p)^T \text{ReLU}(\mathbf{U}\widetilde{\mathbf{H}}^p), \quad \bar{\mathbf{A}} = \text{kNN}(\mathbf{A}), \quad \mathbf{A}^{\dashv}, \mathbf{A}^{\vdash} = \text{softmax}(\{\bar{\mathbf{A}}, \bar{\mathbf{A}}^T\}) \quad (3)$$

where $\mathbf{U}$ is a $d \times (\widetilde{F}_p + \widetilde{F}_a)$ trainable weight matrix. Note that the supervision signal is able to back-propagate through the graph sparsification operation as the $K$ nearest attention scores are kept.

### 2.3.2 BIDIRECTIONAL GATED GRAPH NEURAL NETWORKS

To effectively learn the graph embeddings from the constructed text graph, we propose a novel Bidirectional Gated Graph Neural Network (BiGGNN) which extends Gated Graph Sequence Neural Networks (Li et al., 2015) by learning node embeddings from both incoming and outgoing edges in an interleaved fashion when processing the directed passage graph. Similar idea has also been exploited in (Xu et al., 2018a), which extended another popular variant of GNNs - GraphSAGE (Hamilton et al., 2017). However, one of key difference between our BiGGNN and their bidirectional GraphSAGE is that we fuse the intermediate node embeddings from both incoming and outgoing directions in every iteration, whereas their model simply learns the node embeddings of each direction independently and concatenates them in the final step.

In BiGGNN, node embeddings are initialized to the passage embeddings $\mathbf{X}$ returned by DAN. The same set of network parameters are shared at every hop of computation. At each computation hop, for every node in the graph, we apply an aggregation function which takes as input a set of incoming (or outgoing) neighboring node vectors and outputs a backward (or forward) aggregation vector. For the syntax-based static graph, we use a mean aggregator for simplicity although other operators such as max or attention (Veličković et al., 2017) could also be employed,

$$\begin{aligned} \mathbf{h}^k_{\mathcal{N}_{\dashv(v)}} &= \text{MEAN}(\{\mathbf{h}^{k-1}_v\} \cup \{\mathbf{h}^{k-1}_u, \forall u \in \mathcal{N}_{\dashv(v)}\}) \\ \mathbf{h}^k_{\mathcal{N}_{\vdash(v)}} &= \text{MEAN}(\{\mathbf{h}^{k-1}_v\} \cup \{\mathbf{h}^{k-1}_u, \forall u \in \mathcal{N}_{\vdash(v)}\}) \end{aligned} \quad (4)$$

For the semantics-aware dynamic graph we compute a weighted average for aggregation where the weights come from the normalized adjacency matrices $\mathbf{A}^{\dashv}$ and $\mathbf{A}^{\vdash}$, defined as,

$$\mathbf{h}^k_{\mathcal{N}_{\dashv(v)}} = \sum_{\forall u \in \mathcal{N}_{\dashv(v)}} \mathbf{a}^{\dashv}_{v,u} \mathbf{h}^{k-1}_u, \quad \mathbf{h}^k_{\mathcal{N}_{\vdash(v)}} = \sum_{\forall u \in \mathcal{N}_{\vdash(v)}} \mathbf{a}^{\vdash}_{v,u} \mathbf{h}^{k-1}_u \quad (5)$$

While (Xu et al., 2018a) learn separate node embeddings for both directions independently, we opt to fuse information aggregated in two directions at each hop, which we find works better in general.

$$\mathbf{h}^k_{\mathcal{N}_{(v)}} = \text{Fuse}(\mathbf{h}^k_{\mathcal{N}_{\dashv(v)}}, \mathbf{h}^k_{\mathcal{N}_{\vdash(v)}}) \quad (6)$$

We design the fusion function as a gated sum of two information sources,

$$\text{Fuse}(\mathbf{a}, \mathbf{b}) = \mathbf{z} \odot \mathbf{a} + (1 - \mathbf{z}) \odot \mathbf{b}, \quad \mathbf{z} = \sigma(\mathbf{W}_z[\mathbf{a}; \mathbf{b}; \mathbf{a} \odot \mathbf{b}; \mathbf{a} - \mathbf{b}] + \mathbf{b}_z) \quad (7)$$

where $\odot$ is the component-wise multiplication, $\sigma$ is a sigmoid function, and $\mathbf{z}$ is a gating vector.

Finally, a Gated Recurrent Unit (GRU) (Cho et al., 2014) is used to update the node embeddings by incorporating the aggregation information.

$$\mathbf{h}_v^k = \text{GRU}(\mathbf{h}_v^{k-1}, \mathbf{h}_{\mathcal{N}(v)}^k) \tag{8}$$

After $n$ hops of GNN computation, where $n$ is a hyperparameter, we obtain the final state embedding $\mathbf{h}_v^n$ for node $v$. To compute the graph-level embedding, we first apply a linear projection to the node embeddings, and then apply max-pooling over all node embeddings to get a $d$-dim vector $\mathbf{h}^{\mathcal{G}}$.

### 2.3.3 RNN DECODER

On the decoder side, we adopt the same model architecture as other state-of-the-art Seq2Seq models where an attention-based (Bahdanau et al., 2014; Luong et al., 2015) LSTM decoder with copy (Vinyals et al., 2015; Gu et al., 2016) and coverage mechanisms (Tu et al., 2016) is employed. The decoder takes the graph-level embedding $\mathbf{h}^{\mathcal{G}}$ followed by two separate fully-connected layers as initial hidden states (i.e., $\mathbf{c}_0$ and $\mathbf{s}_0$) and the node embeddings $\{\mathbf{h}_v^n, \forall v \in \mathcal{G}\}$ as the attention memory, and generates the output sequence one word at a time. The particular decoder used in this work closely follows (See et al., 2017). We refer the readers to Appendix A for more details.

### 2.4 HYBRID EVALUATOR

It has been observed that optimizing such cross-entropy based training objectives for sequence learning does not always produce the best results on discrete evaluation metrics (Ranzato et al., 2015; Wu et al., 2016; Paulus et al., 2017). Major limitations of this strategy include exposure bias and evaluation discrepancy between training and testing. To tackle these issues, some recent QG approaches (Song et al., 2017; Kumar et al., 2018b) directly optimize evaluation metrics using REIN-FORCE. We further use a mixed objective function with both syntactic and semantic constraints for guiding text generation. In particular, we present a hybrid evaluator with a mixed objective function that combines both cross-entropy loss and RL loss in order to ensure the generation of syntactically and semantically valid text.

For the RL part, we employ the self-critical sequence training (SCST) algorithm (Rennie et al., 2017) to directly optimize the evaluation metrics. SCST is an efficient REINFORCE algorithm that utilizes the output of its own test-time inference algorithm to normalize the rewards it experiences. In SCST, at each training iteration, the model generates two output sequences: the sampled output $Y^s$, produced by multinomial sampling, that is, each word $y_t^s$ is sampled according to the likelihood $P(y_t|X, y_{<t})$ predicted by the generator, and the baseline output $\hat{Y}$, obtained by greedy search, that is, by maximizing the output probability distribution at each decoding step. We define $r(Y)$ as the reward of an output sequence $Y$, computed by comparing it to corresponding ground-truth sequence $Y^*$ with some reward metrics. The loss function is defined as:

$$\mathcal{L}_{rl} = (r(\hat{Y}) - r(Y^s)) \sum_t \log P(y_t^s|X, y_{<t}^s) \tag{9}$$

As we can see, if the sampled output has a higher reward than the baseline one, we maximize its likelihood, and vice versa.

One of the key factors for RL is to pick the proper reward function. To take syntactic and semantic constraints into account, we consider the following metrics as our reward functions:

*Evaluation metric as reward function:* We use one of our evaluation metrics, BLEU-4, as our reward function $f_{\text{eval}}$, which lets us directly optimize the model towards the evaluation metrics.

*Semantic metric as reward function:* One drawback of some evaluation metrics like BLEU is that they do not measure meaning, but only reward systems that have exact n-gram matches in the reference system. To make our reward function more effective and robust, we additionally use word movers distance (WMD) as a semantic reward function $f_{\text{sem}}$. WMD is the state-of-the-art approach to measure the dissimilarity between two sentences based on word embeddings (Kusner et al., 2015). Following Gong et al. (2019), we take the negative of the WMD distance between a generated sequence and the ground-truth sequence and divide it by the sequence length as its semantic score.

We define the final reward function as $r(Y) = f_{\text{eval}}(Y, Y^*) + \alpha f_{\text{sem}}(Y, Y^*)$ where $\alpha$ is a scalar.

## 2.5 Training and Testing

We train our model in two stages. In the first state, we train the model using regular cross-entropy loss, defined as,

$$\mathcal{L}_{lm} = \sum_t - \log P(y_t^* | X, y_{<t}^*) + \lambda \, \text{covloss}_t \tag{10}$$

where $y_t^*$ is the word at the $t$-th position of the ground-truth output sequence and $\text{covloss}_t$ is the coverage loss defined as $\sum_i min(a_i^t, c_i^t)$, with $a_i^t$ being the $i$-th element of the attention vector over the input sequence at time step $t$. Scheduled teacher forcing (Bengio et al., 2015) is adopted to alleviate the exposure bias problem. In the second stage, we fine-tune the model by optimizing a mixed objective function combining both cross-entropy loss and RL loss, defined as,

$$\mathcal{L} = \gamma \mathcal{L}_{rl} + (1 - \gamma) \mathcal{L}_{lm} \tag{11}$$

where $\gamma$ is a scaling factor controlling the trade-off between cross-entropy loss and RL loss. During the testing phase, we use beam search to generate final predictions.

## 3 Experiments

We evaluate our proposed model against state-of-the-art methods on the SQuAD dataset (Rajpurkar et al., 2016). Our full models have two variants $G2S_{sta}$+BERT+RL and $G2S_{dyn}$+BERT+RL which adopts static graph construction or dynamic graph construction, respectively. For model settings and sensitivity analysis, please refer to Appendix B and C. The implementation of our model is publicly available at `https://github.com/hugochan/RL-based-Graph2Seq-for-NQG`.

### 3.1 Baseline Methods

We compare against the following baselines in our experiments: i) Transformer (Vaswani et al., 2017), ii) SeqCopyNet (Zhou et al., 2018), iii) NQG++ (Zhou et al., 2017), iv) MPQG+R (Song et al., 2017), v) AFPQA (Sun et al., 2018), vi) s2sa-at-mp-gsa (Zhao et al., 2018), vii) ASs2s (Kim et al., 2018), and viii) CGC-QG (Liu et al., 2019). Detailed descriptions of the baselines are provided in Appendix D. Experiments on baselines followed by * are conducted using released code. Results of other baselines are taken from the corresponding papers, with unreported metrics marked as –.

### 3.2 Data and Metrics

SQuAD contains more than 100K questions posed by crowd workers on 536 Wikipedia articles. Since the test set of the original SQuAD is not publicly available, the accessible parts ($\approx$90%) are used as the entire dataset in our experiments. For fair comparison with previous methods, we evaluated our model on both data split-1 (Song et al., 2018a)[1] that contains 75,500/17,934/11,805 (train/development/test) examples and data split-2 (Zhou et al., 2017)[2] that contains 86,635/8,965/8,964 examples.

Following previous works, we use BLEU-4 (Papineni et al., 2002), METEOR (Banerjee & Lavie, 2005), ROUGE-L (Lin, 2004) and Q-BLEU1 (Nema & Khapra, 2018) as our evaluation metrics. Initially, BLEU-4 and METEOR were designed for evaluating machine translation systems and ROUGE-L was designed for evaluating text summarization systems. Recently, Q-BLEU1 was designed for better evaluating question generation systems, which was shown to correlate significantly better with human judgments compared to existing metrics.

Besides automatic evaluation, we also conduct a human evaluation study on split-2. We ask human evaluators to rate generated questions from a set of anonymized competing systems based on whether they are syntactically correct, semantically correct and relevant to the passage. The rating scale is from 1 to 5, on each of the three categories. Evaluation scores from all evaluators are collected and averaged as final scores. Further details on human evaluation can be found in Appendix E.

Table 1: Automatic evaluation results on the SQuAD test set.

| Methods | Split-1 | | | | Split-2 | | | |
|---|---|---|---|---|---|---|---|---|
| | BLEU-4 | METEOR | ROUGE-L | Q-BLEU1 | BLEU-4 | METEOR | ROUGE-L | Q-BLEU1 |
| Transformer | 2.56 | 8.98 | 26.01 | 16.70 | 3.09 | 9.68 | 28.86 | 20.10 |
| SeqCopyNet | – | – | – | – | 13.02 | – | 44.00 | – |
| NQG++ | – | – | – | – | 13.29 | – | – | – |
| MPQG+R* | 14.39 | 18.99 | 42.46 | 52.00 | 14.71 | 18.93 | 42.60 | 50.30 |
| AFPQA | – | – | – | – | 15.64 | – | – | – |
| s2sa-at-mp-gsa | 15.32 | 19.29 | 43.91 | – | 15.82 | 19.67 | 44.24 | – |
| ASs2s | 16.20 | 19.92 | 43.96 | – | 16.17 | – | – | – |
| CGC-QG | – | – | – | – | 17.55 | 21.24 | 44.53 | – |
| G2S$_{dyn}$+BERT+RL | 17.55 | 21.42 | 45.59 | 55.40 | 18.06 | 21.53 | 45.91 | 55.00 |
| G2S$_{sta}$+BERT+RL | **17.94** | **21.76** | **46.02** | **55.60** | **18.30** | **21.70** | **45.98** | **55.20** |

Table 2: Human evaluation results ($\pm$ standard deviation) on the SQuAD split-2 test set. The rating scale is from 1 to 5 (higher scores indicate better results).

| Methods | Syntactically correct | Semantically correct | Relevant |
|---|---|---|---|
| MPQG+R* | 4.34 (0.15) | 4.01 (0.23) | 3.21 (0.31) |
| G2S$_{sta}$+BERT+RL | 4.41 (0.09) | 4.31 (0.12) | 3.79 (0.45) |
| Ground-truth | **4.74 (0.14)** | **4.74 (0.19)** | **4.25 (0.38)** |

## 3.3 EXPERIMENTAL RESULTS AND HUMAN EVALUATION

Table 1 shows the automatic evaluation results comparing our proposed models against other state-of-the-art baseline methods. First of all, we can see that both of our full models G2S$_{sta}$+BERT+RL and G2S$_{dyn}$+BERT+RL achieve the new state-of-the-art scores on both data splits and consistently outperform previous methods by a significant margin. This highlights that our RL-based Graph2Seq model, together with the deep alignment network, successfully addresses the three issues we highlighted in Sec. 1. Between these two variants, G2S$_{sta}$+BERT+RL outperforms G2S$_{dyn}$+BERT+RL on all the metrics. Also, unlike the baseline methods, our model does not rely on any hand-crafted rules or ad-hoc strategies, and is fully end-to-end trainable.

As shown in Table 2, we conducted a human evaluation study to assess the quality of the questions generated by our model, the baseline method MPQG+R, and the ground-truth data in terms of syntax, semantics and relevance metrics. We can see that our best performing model achieves good results even compared to the ground-truth, and outperforms the strong baseline method MPQG+R. Our error analysis shows that main syntactic error occurs in repeated/unknown words in generated questions. Further, the slightly lower quality on semantics also impacts the relevance.

## 3.4 ABLATION STUDY

Table 3: Ablation study on the SQuAD split-2 test set.

| Methods | BLEU-4 | Methods | BLEU-4 |
|---|---|---|---|
| G2S$_{dyn}$+BERT+RL | 18.06 | G2S$_{dyn}$ | 16.81 |
| G2S$_{sta}$+BERT+RL | 18.30 | G2S$_{sta}$ | 16.96 |
| G2S$_{sta}$+BERT-fixed+RL | 18.20 | G2S$_{dyn}$ w/o DAN | 12.58 |
| G2S$_{dyn}$+BERT | 17.56 | G2S$_{sta}$ w/o DAN | 12.62 |
| G2S$_{sta}$+BERT | 18.02 | G2S$_{sta}$ w/o BiGGNN, w/ Seq2Seq | 16.14 |
| G2S$_{sta}$+BERT-fixed | 17.86 | G2S$_{sta}$ w/o BiGGNN, w/ GCN | 14.47 |
| G2S$_{dyn}$+RL | 17.18 | G2S$_{sta}$ w/ GGNN-forward | 16.53 |
| G2S$_{sta}$+RL | 17.49 | G2S$_{sta}$ w/ GGNN-backward | 16.75 |

---

[1]`https://www.cs.rochester.edu/~lsong10/downloads/nqg_data.tgz`
[2]`https://res.qyzhou.me/redistribute.zip`

As shown in Table 3, we perform an ablation study to systematically assess the impact of different model components (e.g., BERT, RL, DAN, and BiGGNN) for two proposed full model variants (static vs dynamic) on the SQuAD split-2 test set. It confirms our finding that syntax-based static graph construction (G2S$_{sta}$+BERT+RL) performs better than semantics-aware dynamic graph construction (G2S$_{dyn}$+BERT+RL) in almost every setting. However, it may be too early to conclude which one is the method of choice for QG. On the one hand, an advantage of static graph construction is that useful domain knowledge can be hard-coded into the graph, which can greatly benefit the downstream task. However, it might suffer if there is a lack of prior knowledge for a specific domain knowledge. On the other hand, dynamic graph construction does not need any prior knowledge about the hidden structure of text, and only relies on the attention matrix to capture these structured information, which provides an easy way to achieve a decent performance. One interesting direction is to explore effective ways of combining both static and dynamic graphs.

By turning off the Deep Alignment Network (DAN), the BLEU-4 score of G2S$_{sta}$ (similarly for G2S$_{dyn}$) dramatically drops from $16.96\%$ to $12.62\%$, which indicates the importance of answer information for QG and shows the effectiveness of DAN. This can also be verified by comparing the performance between the DAN-enhanced Seq2Seq model (16.14 BLEU-4 score) and other carefully designed answer-aware Seq2Seq baselines such as NQG++ (13.29 BLEU-4 score), MPQG+R (14.71 BLEU-4 score) and AFPQA (15.82 BLEU-4 score). Further experiments demonstrate that both word-level (G2S$_{sta}$ w/ DAN-word only) and contextual-level (G2S$_{sta}$ w/ DAN-contextual only) answer alignments in DAN are helpful.

We can see the advantages of Graph2Seq learning over Seq2Seq learning on this task by comparing the performance between G2S$_{sta}$ and Seq2Seq. Compared to Seq2Seq based QG methods that completely ignore hidden structure information in the passage, our Graph2Seq based method is aware of more hidden structure information such as semantic similarity between any pair of words that are not directly connected or syntactic relationships between two words captured in a dependency parsing tree. In our experiments, we also observe that doing both forward and backward message passing in the GNN encoder is beneficial. Surprisingly, using GCN (Kipf & Welling, 2016) as the graph encoder (and converting the input graph to an undirected graph) does not provide good performance. In addition, fine-tuning the model using REINFORCE can further improve the model performance in all settings (i.e., w/ and w/o BERT), which shows the benefits of directly optimizing the evaluation metrics. Besides, we find that the pretrained BERT embedding has a considerable impact on the performance and fine-tuning BERT embedding even further improves the performance, which demonstrates the power of large-scale pretrained language models.

## 3.5 CASE STUDY

Table 4: Generated questions on SQuAD split-2 test set. Target answers are underlined.

| |
|---|
| **Passage:** for the successful execution of a project , effective planning is essential . |
| **Gold:** what is essential for the successful execution of a project ? |
| **G2S$_{sta}$ w/o BiGGNN (Seq2Seq):** what type of planning is essential for the project ? |
| **G2S$_{sta}$ w/o DAN.:** what type of planning is essential for the successful execution of a project ? |
| **G2S$_{sta}$:** what is essential for the successful execution of a project ? |
| **G2S$_{sta}$+BERT:** what is essential for the successful execution of a project ? |
| **G2S$_{sta}$+BERT+RL:** what is essential for the successful execution of a project ? |
| **G2S$_{dyn}$+BERT+RL:** what is essential for the successful execution of a project ? |
| **Passage:** the church operates three hundred sixty schools and institutions overseas . |
| **Gold:** how many schools and institutions does the church operate overseas ? |
| **G2S$_{sta}$ w/o BiGGNN (Seq2Seq):** how many schools does the church have ? |
| **G2S$_{sta}$ w/o DAN.:** how many schools does the church have ? |
| **G2S$_{sta}$:** how many schools and institutions does the church have ? |
| **G2S$_{sta}$+BERT:** how many schools and institutions does the church have ? |
| **G2S$_{sta}$+BERT+RL:** how many schools and institutions does the church operate ? |
| **G2S$_{dyn}$+BERT+RL:** how many schools does the church operate ? |

In Table 4, we further show a few examples that illustrate the quality of generated text given a passage under different ablated systems. As we can see, incorporating answer information helps the

model identify the answer type of the question to be generated, and thus makes the generated questions more relevant and specific. Also, we find our Graph2Seq model can generate more complete and valid questions compared to the Seq2Seq baseline. We think it is because a Graph2Seq model is able to exploit the rich text structure information better than a Seq2Seq model. Lastly, it shows that fine-tuning the model using REINFORCE can improve the quality of the generated questions.

# 4 RELATED WORK

## 4.1 NATURAL QUESTION GENERATION

Early works (Mostow & Chen, 2009; Heilman & Smith, 2010) for QG focused on rule-based approaches that rely on heuristic rules or hand-crafted templates, with low generalizability and scalability. Recent attempts have focused on NN-based approaches that do not require manually-designed rules and are end-to-end trainable. Existing NN-based approaches (Du et al., 2017; Yao et al.; Zhou et al., 2018) rely on the Seq2Seq model with attention, copy or coverage mechanisms. In addition, various ways (Zhou et al., 2017; Song et al., 2017; Zhao et al., 2018) have been proposed to utilize the target answer for guiding the question generation. Some recent approaches (Song et al., 2017; Kumar et al., 2018b) aim at directly optimizing evaluation metrics using REINFORCE. Concurrent works have explored tackling the QG task with various semantics-enhanced rewards (Zhang & Bansal, 2019) or large-scale pretrained language models (Dong et al., 2019).

However, the existing approaches for QG suffer from several limitations; they (i) ignore the rich structure information hidden in text, (ii) solely rely on cross-entropy loss that leads to issues like exposure bias and inconsistency between train/test measurement, and (iii) fail to fully exploit the answer information. To address these limitations, we propose a RL based Graph2Seq model augmented with a deep alignment network to effectively tackle the QG task. To the best of our knowledge, we are the first to introduce the Graph2Seq architecture to solve the question generation task.

## 4.2 GRAPH NEURAL NETWORKS

Over the past few years, graph neural networks (GNNs) (Kipf & Welling, 2016; Gilmer et al., 2017; Hamilton et al., 2017) have attracted increasing attention. Due to more recent advances in graph representation learning, a number of works have extended the widely used Seq2Seq architectures (Sutskever et al., 2014; Cho et al., 2014) to Graph2Seq architectures for machine translation, semantic parsing, AMR(SQL)-to-text, and online forums health stage prediction tasks (Bastings et al., 2017; Beck et al., 2018; Xu et al., 2018a;b;c; Song et al., 2018b; Gao et al., 2019). While the high-quality graph structure is crucial for the performance of GNN-based approaches, most existing works use syntax-based static graph structures when applied to textual data. Very recently, researchers have started exploring methods to automatically construct a graph of visual objects (Norcliffe-Brown et al., 2018) or words (Liu et al., 2018; Chen et al., 2019c;b) when applying GNNs to non-graph structured data. To the best of our knowledge, we are the first to investigate systematically the performance difference between syntactic-aware static graph construction and semantics-aware dynamic graph construction in the context of question generation.

# 5 CONCLUSION

We proposed a novel RL based Graph2Seq model for QG, where the answer information is utilized by an effective Deep Alignment Network and a novel bidirectional GNN is proposed to process the directed passage graph. On the SQuAD dataset, our method outperforms existing methods by a significant margin and achieves the new state-of-the-art results. Future directions include investigating more effective ways of automatically learning graph structures from text and exploiting Graph2Seq models for question generation from structured data like knowledge graphs or tables.

## ACKNOWLEDGMENTS

This work is supported by IBM Research AI through the IBM AI Horizons Network. We thank the human evaluators who evaluated our system. We also thank the anonymous reviewers for their constructive feedback.

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

# A    DETAILS ON THE RNN DECODER

At each decoding step $t$, an attention mechanism learns to attend to the most relevant words in the input sequence, and computes a context vector $\mathbf{h}_t^*$ based on the current decoding state $\mathbf{s}_t$, the current coverage vector $\mathbf{c}^t$ and the attention memory. In addition, the generation probability $p_{\text{gen}} \in [0, 1]$ is calculated from the context vector $\mathbf{h}_t^*$, the decoder state $\mathbf{s}_t$ and the decoder input $y_{t-1}$. Next, $p_{\text{gen}}$ is used as a soft switch to choose between generating a word from the vocabulary, or copying a word from the input sequence. We dynamically maintain an extended vocabulary which is the union of the usual vocabulary and all words appearing in a batch of source examples (i.e., passages and answers). Finally, in order to encourage the decoder to utilize the diverse components of the input sequence, a coverage mechanism is applied. At each step, we maintain a coverage vector $\mathbf{c}^t$, which is the sum of attention distributions over all previous decoder time steps. A coverage loss is also computed to penalize repeatedly attending to the same locations of the input sequence.

# B    MODEL SETTINGS

We keep and fix the 300-dim GloVe vectors for the most frequent 70,000 words in the training set. We compute the 1024-dim BERT embeddings on the fly for each word in text using a (trainable) weighted sum of all BERT layer outputs. The embedding sizes of case, POS and NER tags are set to 3, 12 and 8, respectively. We set the hidden state size of BiLSTM to 150 so that the concatenated state size for both directions is 300. The size of all other hidden layers is set to 300. We apply a variational dropout (Kingma et al., 2015) rate of 0.4 after word embedding layers and 0.3 after RNN layers. We set the neighborhood size to 10 for dynamic graph construction. The number of GNN hops is set to 3. During training, in each epoch, we set the initial teacher forcing probability to 0.75 and exponentially increase it to $0.75 * 0.9999^i$ where $i$ is the training step. We set $\alpha$ in the reward function to 0.1, $\gamma$ in the mixed loss function to 0.99, and the coverage loss ratio $\lambda$ to 0.4. We use Adam (Kingma & Ba, 2014) as the optimizer, and the learning rate is set to 0.001 in the pretraining stage and 0.00001 in the fine-tuning stage. We reduce the learning rate by a factor of 0.5 if the validation BLEU-4 score stops improving for three epochs. We stop the training when no improvement is seen for 10 epochs. We clip the gradient at length 10. The batch size is set to 60 and 50 on data split-1 and split-2, respectively. The beam search width is set to 5. All hyperparameters are tuned on the development set.

# C    SENSITIVITY ANALYSIS OF HYPERPARAMETERS

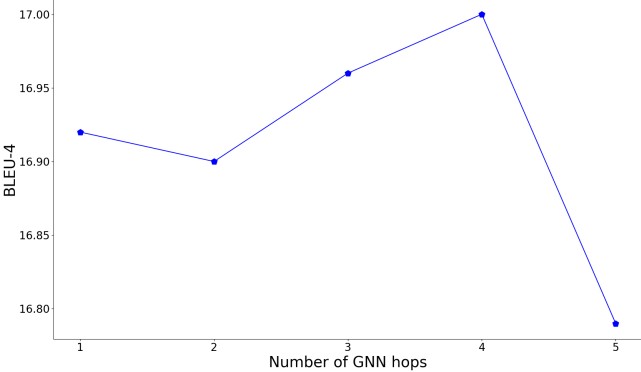

Figure 3: Effect of the number of GNN hops.

To study the effect of the number of GNN hops, we conduct experiments on the G2S$_{sta}$ model on the SQuAD split-2 data. Fig. 3 shows that our model is not very sensitive to the number of GNN hops and can achieve reasonably good results with various number of hops.

Table 5: Ablation study on the SQuAD split-2 test set.

| Methods | BLEU-4 | Methods | BLEU-4 |
|---|---|---|---|
| G2S$_{dyn}$+BERT+RL | 18.06 | G2S$_{dyn}$ w/o feat | 16.51 |
| G2S$_{sta}$+BERT+RL | 18.30 | G2S$_{sta}$ w/o feat | 16.65 |
| G2S$_{sta}$+BERT-fixed+RL | 18.20 | G2S$_{dyn}$ w/o DAN | 12.58 |
| G2S$_{dyn}$+BERT | 17.56 | G2S$_{sta}$ w/o DAN | 12.62 |
| G2S$_{sta}$+BERT | 18.02 | G2S$_{sta}$ w/ DAN-word only | 15.92 |
| G2S$_{sta}$+BERT-fixed | 17.86 | G2S$_{sta}$ w/ DAN-contextual only | 16.07 |
| G2S$_{dyn}$+RL | 17.18 | G2S$_{sta}$ w/ GGNN-forward | 16.53 |
| G2S$_{sta}$+RL | 17.49 | G2S$_{sta}$ w/ GGNN-backward | 16.75 |
| G2S$_{dyn}$ | 16.81 | G2S$_{sta}$ w/o BiGGNN, w/ Seq2Seq | 16.14 |
| G2S$_{sta}$ | 16.96 | G2S$_{sta}$ w/o BiGGNN, w/ GCN | 14.47 |

# D    DETAILS ON BASELINE METHODS

**Transformer** (Vaswani et al., 2017) We included a Transformer-based Seq2Seq model augmented with attention and copy mechanisms. We used the open source implementation[3] provided by the OpenNMT (Klein et al., 2017) library and trained the model from scratch. Surprisingly, this baseline performed very poorly on the benchmarks even though we conducted moderate hyperparameter search and trained the model for a large amount of epochs. We suspect this might be partially because this method is very sensitive to hyperparameters as reported by Klein et al. (2017) and probably data-hungry on this task. We conjecture that better performance might be expected by extensively searching the hyperparameters and using a pretrained transformer model.

**SeqCopyNet** (Zhou et al., 2018) proposed an extension to the copy mechanism which learns to copy not only single words but also sequences from the input sentence.

**NQG++** (Zhou et al., 2017) proposed an attention-based Seq2Seq model equipped with a copy mechanism and a feature-rich encoder to encode answer position, POS and NER tag information.

**MPQG+R** (Song et al., 2017) proposed an RL-based Seq2Seq model with a multi-perspective matching encoder to incorporate answer information. Copy and coverage mechanisms are applied.

**AFPQA** (Sun et al., 2018) consists of an answer-focused component which generates an interrogative word matching the answer type, and a position-aware component which is aware of the position of the context words when generating a question by modeling the relative distance between the context words and the answer.

**s2sa-at-mp-gsa** (Zhao et al., 2018) proposed a model which contains a gated attention encoder and a maxout pointer decoder to tackle the challenges of processing long input sequences. For fair comparison, we report the results of the sentence-level version of their model to match with our settings.

**ASs2s** (Kim et al., 2018) proposed an answer-separated Seq2Seq model which treats the passage and the answer separately.

**CGC-QG** (Liu et al., 2019) proposed a multi-task learning framework to guide the model to learn the accurate boundaries between copying and generation.

# E    DETAILS ON HUMAN EVALUATION

We conducted a small-scale (i.e., 50 random examples per system) human evaluation on the split-2 data. We asked 5 human evaluators to give feedback on the quality of questions generated by a set of anonymized competing systems. In each example, given a triple containing a source passage, a target answer and an anonymised system output, they were asked to rate the quality of the output by answering the following three questions: i) is this generated question syntactically correct? ii) is this generated question semantically correct? and iii) is this generated question relevant to the passage? For each evaluation question, the rating scale is from 1 to 5 where a higher score means

---

[3] https://opennmt.net/OpenNMT-py/FAQ.html

better quality (i.e., 1: Poor, 2: Marginal, 3: Acceptable, 4: Good, 5: Excellent). Responses from all evaluators were collected and averaged.

# F  MORE RESULTS ON ABLATION STUDY

We perform the comprehensive ablation study to systematically assess the impact of different model components (e.g., BERT, RL, DAN, BiGGNN, FEAT, DAN-word, and DAN-contextual) for two proposed full model variants (static vs dynamic) on the SQuAD split-2 test set. Our experimental results confirmed that every component in our proposed model makes the contribution to the overall performance.

