# OpenReview forum: "Reinforcement Learning Based Graph-to-Sequence Model for Natural Question Generation"
_ICLR.cc/2020/Conference — Accept (Poster)_

### Official Review · AnonReviewer3 · 2019-10-12
**Official Blind Review #3**

**Rating:** 8

**Review:**

The authors propose a Graph-to-Sequence Reinforcement Learning Model for Natural Question Generation, evaluated on SQuAD benchmark in for Question Generation. An interesting aspect of the work is related to the Graph2Seq model, and the use of the Reinforcement Learning to fine-tune the model. The latter stage seems to improve the structure of the answers considerably. An interesting use of RL algorithm and apparently a good choice of reward functions.

Questions: in the combined loss used in the RL run:
1. Have you managed to have a successful run with gamma = 1?
2. I understand that the L_rl factor is computed based on the sampling, and the L_lm is computed based on the top variant from the nbest list?



**Experience Assessment:**

I have published one or two papers in this area.

**Review Assessment: Checking Correctness Of Derivations And Theory:**

I did not assess the derivations or theory.

**Review Assessment: Checking Correctness Of Experiments:**

I did not assess the experiments.

**Review Assessment: Thoroughness In Paper Reading:**

I made a quick assessment of this paper.

---

> ### Author Response · Authors · 2019-11-11
> **Author response to Review #3**
>
> We are very grateful to the reviewer for this accurate summary, and for the kind recognition of our key contributions.
> Below we address the concerns mentioned in the review:
>
> 1) Have you managed to have a successful run with gamma = 1?
>
> In our experiments, we set gamma to 0.99 which is already very close to 1. We found that using even larger gamma slightly harmed the model performance. We will add discussion of the effect of gamma.
>
> 2) I understand that the L_rl factor is computed based on the sampling, and the L_lm is computed based on the top variant from the nbest list?
>
> When computing the L_lm, we use the ground-truth output sequence as supervision.
> Scheduled teacher forcing is adopted to alleviate the exposure bias problem, which means we randomly choose to either use the ground-truth token or the best token predicted at last step as the input to the current step of computation. We will make it more clear in the revision.

---

### Official Review · AnonReviewer2 · 2019-10-17
**Official Blind Review #2**

**Rating:** 6

**Review:**

This paper focuses on improving the performance on the task of natural language generation. To this end, they propose a  graph-to-sequence (Graph2Seq) model for the task of question generation which exploits the rich structure information in the text as well as use reinforcement learning based policy gradient approach to address the exposure bias and inconsistency between test/train distributions in cross-entropy optimization setup.

The Graph2Seq model has a bidirectional gated graph neural network on the encoder side, which is an extension of traditional gated graph neural network. To exploit the rich hidden structure information in the input text, they explore two different methods: (1) syntax-based static graph; (2) semantics-aware dynamics graph.

The proposed model achieves state-of-the-art results on question generation, which are further validated with human evaluations.

Overall, The paper should be rejected because the paper have minor extensions to each of their modules but lacking any major important contribution.
Some major concerns:
1) The bidirectional gated GNN doesn’t seem novel enough in comparison to previous work
2) I believe RL to Graph2Seq is a minor extension from Seq2Seq, since RL mostly deals with the decoder part which is common in across both Graph2Seq and Seq2Seq


Arguments:

1) Adding the structure information to the encoder via the GNNs is an interesting angle for question generation. Compared to previous work, this paper proposes an additional deep alignment network on the encoder side to align paragraph and answer. However, the importance of this module is not well studied in the experiments section. I see that there is an ablation with/without this module but its not fairly compared with other aligning or simple techniques like in Zhao et al. (2018).

2) The addition of RL component to Graph2Seq is a minor extension from the Seq2Seq model, because both of these models have similar decoder and RL mainly deals with it. Also the importance of each reward component or the effect of each phrase-matching automatic metrics is missing.

3) Open part I am unclear about the dataset is which dataset version did you use sentence-level or paragraph level? I see that the baselines correspond to sentence-level, but the Figure-1 alignment module has input paragraph. Also I couldn’t find the SeqCopyNet (Zhou et al., 2018) split-2 BLEU4 score=13.02 in the original paper!

4) Some of the latest papers which use BERT based models are not discussed in the paper which achieve state-of-the-art-results: “Addressing Semantic Drift in Question Generation for Semi-Supervised Question Answering”

5) For Table-2 results are the differences in the scores for the two models statistically significant?

6) Table-3: First of all, evaluating only one metric is no sufficient. Please see latest papers that have also introduced new metrics that are good for QG evaluation, e.g., Q-BLEU. The gap between G2Ssta+BERT vs. G2Ssta+BERT+RL seems negligible, and missing statistical significance.

7) Minor comments: BLUE -> BLEU; please cite KNN-style graph sparsification; the color choices in Figure-3 are creating confusion in understanding the model.


**Experience Assessment:**

I have published one or two papers in this area.

**Review Assessment: Checking Correctness Of Derivations And Theory:**

I carefully checked the derivations and theory.

**Review Assessment: Checking Correctness Of Experiments:**

I carefully checked the experiments.

**Review Assessment: Thoroughness In Paper Reading:**

I read the paper thoroughly.

---

> ### Author Response · Authors · 2019-11-11
> **Author response to Review #2**
>
> We want to thank the reviewer for their careful reading and providing a lot of critical comments! Below we address the concerns mentioned in the review:
>
> 1) Overall, The paper should be rejected because the paper have minor extensions to each of their modules but lacking any major important contribution.
>
> Our most important contribution is to present a novel RL-based Graph2Seq model for an important NLP task - question generation (QG), to effectively solve three severe issues with existing approaches: 1) failure to consider global interactions between answer and context (solved by Deep alignment network) ; 2) failure to consider rich hidden structure information of word sequence (solved by Graph2Seq); 3) limitations of cross-entropy based objectives (solved by RL). We believe that the combination of the above techniques for solving the QG task is novel and NLP researchers working on QG will find our approach beneficial. We further explore both static and dynamic approaches for constructing graphs when applying GNNs to textual data. To the best of our knowledge, this kind of empirical comparison has not been conducted in previous works.
>
> 2a) The importance of this module (deep answer alignment) is not well studied in the experiments section. It’s not fairly compared with other aligning or simple techniques like in Zhao et al. (2018).
>
> A Deep Alignment Network (DAN) is proposed to effectively incorporate answer information into the passage with multiple granularity levels (word level and contextualized hidden state level) with the assumption that multiple levels of alignments can help learn hierarchical representations. In order to study the effectiveness of DAN, we conducted extensive ablation experiments on Squad split 2 data (detailed results are provided in appendix F table 5). We copy some results here:
>
> Models | BLEU-4
> G2S_dyn | 16.81
> G2S_dyn w/o DAN | 12.58
> G2S_sta | 16.96
> G2S_sta w/o DAN | 12.62
> G2S_sta w/ DAN-word only | 15.92
> G2S_sta w/ DAN-hidden only | 16.07
>
> Seq2Seq w/ DAN | 16.14
> Answer-focused Position-aware model | 15.82
> MPQG+R | 14.71
> NQG++ | 13.29
>
> By turning off DAN, the BLEU-4 score of G2S_sta (similarly for G2S_dyn) dramatically drops from 16.96 to 12.62, which shows the effectiveness of DAN. It is also interesting to see that our DAN-enhanced Seq2Seq baseline (16.14 BLEU-4) significantly outperforms other carefully designed answer-aware Seq2Seq baselines such as Answer-focused Position-aware model (15.82 BLEU-4), MPQG+R (14.71 BLEU-4) and NQG++ (13.29 BLEU-4). This demonstrates that the proposed DAN network is effective in general, and not only for the Graph2Seq model. In addition, this also indicates that, as an answer alignment technique, DAN is more effective than previously proposed techniques such as answer position features (used by NQG++ and Answer-focused Position-aware) and multi-perspective matching (used by MPQG+R). Further experiments demonstrate that both word-level (G2S_sta w/ DAN-word only) and hidden-level (G2S_sta w/ DAN-hidden only) answer alignments in DAN are helpful.
>
> 2b) The addition of RL component to Graph2Seq is a minor extension from the Seq2Seq model. The importance of each reward component for RL.
>
> While we are not the first to apply RL to QG models to tackle the issues with cross-entropy based sequence training, it is important to note that unlike existing RL based approaches that usually only employ evaluation metrics like BLEU as rewards for RL optimization, we propose to use both syntactic (e.g., BLEU) and semantic (e.g., word mover’s distance (WMD)) rewards for guiding high-quality text generation.
>
> For evaluating the reward components, as suggested, we did extra experiments to study the impact of different reward components on Squad split 2. The results are as follows:
>
> G2S_sta: 16.96, G2S_sta + RL: 17.49,  G2S_sta + RL w/ BLEU only: 17.30,
> G2S_sta + RL w/ WMD only: 17.14.
>
> It seems that both syntactic and semantic rewards help the model performance.

---

> ### Author Response · Authors · 2019-11-11
> **Author response to Review #2 (continued)**
>
> 3) I am unclear about the dataset is which dataset version did you use sentence-level or paragraph level? I see that the baselines correspond to sentence-level, but the Figure-1 alignment module has input paragraph. Also I couldn’t find the SeqCopyNet (Zhou et al., 2018) split-2 BLEU4 score=13.02 in the original paper!
>
> As mentioned in Sec. 3.2, following previous work, we used two versions of Squad datasets, namely, Squad split 1 and Squad split 2. Squad split 1 was released by Song  et  al.,  2018a, and is available at https://www.cs.rochester.edu/~lsong10/downloads/nqg_data.tgz. Squad split 2 was released by Zhou et al., 2017, and is available at https://res.qyzhou.me/redistribute.zip. Both datasets use sentence-level input, but in our paper, we simply term an input sequence as a passage regardless of the length of the sequence.
>
> In Table 4 of the SeqCopyNet (Zhou et al., 2018) paper,  you should be able to find the BLEU-4 score on Squad split 2 which is 13.02. Here is a link to the paper: https://arxiv.org/pdf/1807.02301.pdf.
>
> 4) Some of the latest papers which use BERT based models are not discussed in the paper which achieve state-of-the-art-results: “Addressing Semantic Drift in Question Generation for Semi-Supervised Question Answering”
>
> Thank you for referring us to the paper. It seems this paper first appeared on arXiv on Sep 13, 2019, which was close to the ICLR2020 submission deadline. So we were not aware of this paper. We will discuss this paper and potentially other BERT based models in our revision.
>
> 5) For Table-2 results are the differences in the scores for the two models statistically significant?
>
> That’s a good suggestion! We noted that previous works on QG also only report average and/or standard deviation on the human evaluation study. Following previous work, we in addition report the standard deviation together with the average score for the human evaluation study as below. We can see that our best performing model achieves good results even compared to the ground-truth, and outperforms the strong baseline method MPQG+R.
>
> Methods | syntactic | semantic | relevance
> MPQG+R: 4.34 (0.15) | 4.01 (0.23) | 3.21 (0.31)
> G2S_sta+BERT+RL: 4.41 (0.09) | 4.31 (0.12) | 3.79 (0.45)
> Ground-truth: 4.74 (0.14) | 4.74 (0.19) | 4.25 (0.38)
>
> 6) Table-3: First of all, evaluating only one metric is no sufficient. Please see latest papers that have also introduced new metrics that are good for QG evaluation, e.g., Q-BLEU. The gap between G2Ssta+BERT vs. G2Ssta+BERT+RL seems negligible, and missing statistical significance.
>
> Following the prior works, we reported BLEU-4, METEOR and ROUGE-L on Squad split 1, and only BLEU-4 on Squad split 2. However, as suggested by the reviewer, below are the results of other evaluation metrics for our model and MPQG+R baseline (we ran their code on our own):
>
> Squad split 1
> Methods \ Metrics | BLEU-4 | METEOR | ROUGE-L | Q-BLEU1
> Transformer-based seq2seq | 2.63 | 7.13 | 24.20 | 15.1
> MPQG+R (Song et al., 2017) | 14.39 | 18.99 | 42.46 | 52.0
> G2S_dyn + BERT + RL | 17.55 | 21.42 | 45.59 | 55.4
> G2S_sta + BERT + RL | 17.94 | 21.76 | 46.02 | 55.6
>
> Squad split 2
> Methods \ Metrics | BLEU-4 | METEOR | ROUGE-L | Q-BLEU1
> Transformer-based seq2seq | 2.52 | 7.25 | 25.77 | 18.0
> NQG (Zhou et al., 2017) | 12.23 | 17.91 | 40.29 | 49.8
> MPQG+R (Song et al., 2017) | 14.71 | 18.93 | 42.6 | 50.3
> G2S_dyn + BERT + RL | 18.06 | 21.53 | 45.91 | 55.0
> G2S_sta + BERT + RL | 18.30 | 21.70 | 45.98 | 55.2
>
> In all cases, our model G2S_sta + BERT + RL yields the best results. For transformer-based Seq2Seq see response to review #1. The Q-BLEU1 was computed using the script available at https://github.com/PrekshaNema25/Answerability-Metric with the following command suggested in the original paper:
>
> python answerability_score.py --data_type squad --ref_file references.txt --hyp_file hypotheses.txt --ner_weight 0.41 --qt_weight 0.2 --re_weight 0.36 --delta 0.66 --ngram_metric Bleu_1
>
> In our experiments, we found that applying RL to the graph2seq model consistently helps improve the model performance, as shown below:
>
> BLEU-4 on Squad split 2:
> G2S_sta: 16.96 vs G2S_sta+RL: 17.49
> G2S_sta+BERT: 18.02 vs G2S_sta+BERT+RL: 18.30
> G2S_dyn: 16.81 vs G2S_dyn+RL: 17.18
> G2S_dyn+BERT: 17.56 vs G2S_dyn+BERT+RL: 18.06
>
>
> 7) Minor comments: BLUE -> BLEU; please cite KNN-style graph sparsification; the color choices in Figure-3 are creating confusion in understanding the model.
>
> We have fixed the typos and cited relevant works on KNN-style graph sparsification in the updated manuscript. We are not sure why the choices in Figure-3 lead to some confusion. Could you please clarify it in more details?

---

### Official Review · AnonReviewer1 · 2019-10-23
**Official Blind Review #1**

**Rating:** 6

**Review:**

The paper proposes two modules to improve the performance of the Natural Question Generation task: (1) deep alignment network and (2) passage graph embeddings. The idea of generating passage graph is novel. The authors experiment with SQuAD and the numbers look good.

I have a few questions regarding the model and experiments.
First, a reasonable baseline could be using Transformer-based sequence to sequence model. Could you fine tune the embedding of CLS token and use that as a summary of the document? It seems that the construction of the passage graph is basically sparsifying a multi-head attention in the BERT model. I think you should justify why graph-structure is important in your experiment.

Second, if the Graph2Seq is particularly important for Natural Question Generation, the author should clarify it more. If the Graph2Seq model is generally applicable to replace the Seq2Seq model, the author should experiment with more tasks. The paper seems not well motivated.


**Experience Assessment:**

I have read many papers in this area.

**Review Assessment: Checking Correctness Of Derivations And Theory:**

N/A

**Review Assessment: Checking Correctness Of Experiments:**

I assessed the sensibility of the experiments.

**Review Assessment: Thoroughness In Paper Reading:**

I read the paper at least twice and used my best judgement in assessing the paper.

---

> ### Author Response · Authors · 2019-11-11
> **Author response to Review #1**
>
> First of all, we want to thank the reviewer for their thorough reading and valuable comments! However, there are some points of misunderstanding that we address in this rebuttal.
>
> In this work, we proposed RL-based Graph2Seq model for an important NLP task - question generation (QG), to effectively solve three severe issues with existing approaches: 1) failure to consider global interactions between answer and context (solved by Deep alignment network) ; 2) failure to consider rich hidden structure information of word sequence (solved by Graph2Seq); 3) limitations of cross-entropy based objectives (solved by RL). We believe that the combination of the above techniques for solving the QG task is novel and NLP researchers working on QG will find our approach beneficial. We further explore both static and dynamic approaches for constructing graphs when applying GNNs to textual data. To the best of our knowledge, this kind of empirical comparison has not been conducted in previous works.
>
> Below we address the concerns mentioned in the review:
>
> 1) A reasonable baseline could be using Transformer-based sequence to sequence model.
>
> Based on the suggestion, we used the open source Transformer implementation available at https://github.com/OpenNMT/OpenNMT-py with the hyperparameters suggested at http://opennmt.net/OpenNMT-py/FAQ.html#how-do-i-use-the-transformer-model.
> The transformer based seq2seq baseline takes the passage as input, and outputs the question. After training the model for more than 130 epochs, we evaluated the best saved model on the test set. The results of transformer-based seq2seq model are as follows:
>
> Squad split 1
> Methods \ Metrics | BLEU-4 | METEOR | ROUGE-L | Q-BLEU1
> Transformer-based seq2seq | 2.63 | 7.13 | 24.20 | 15.1
>
> Squad split 2
> Methods \ Metrics | BLEU-4 | METEOR | ROUGE-L | Q-BLEU1
> Transformer-based seq2seq | 2.52 | 7.25 | 25.77 | 18.0
>
> Surprisingly, the transformer-based seq2seq performed very poorly even though we have followed the suggested hyperparameters as shown in the above link. We are not sure what’s the possible reasons yet. But we will try to perform more extensive hyperparameter search after the rebuttal and add this baseline in our final version. Nevertheless, our proposed model achieved the BLEU-4 score 17.94 on split1, and 18.30 on split2, which are significantly better than the transformer-based seq2seq baseline. And even our Graph2Seq variant without answer information achieved the BLEU-4 score 12.64 on split2 as shown in Table 3.
>
> 2) It seems that the construction of the passage graph is basically sparsifying a multi-head attention in the BERT model. I think you should justify why graph-structure is important in your experiment.
>
> Yes, this is what we have done for constructing semantic-aware dynamic passage graph (although we just used self-attention instead of multi-head attention). Another type of passage graph we explored is called syntax-based static passage graph, which is constructed by combining the dependency parsing tree with the word sequence. The reason the graph-structure of inputs are important is because it carries more hidden structure information such as semantic similarity between any pair of words that are not directly connected or syntactic relationships between two words captured in a dependency parsing tree. Obviously, existing works using seq2seq for QG completely ignored these hidden structure information. This motivated us to develop Graph2Seq to better leverage these additional information beyond the word sequence. We have discussed them in detail in Sec. 2.3.1 and we have added more discussions in experiments in Sec. 3.4 in the updated manuscript.

---

> ### Author Response · Authors · 2019-11-11
> **Author response to Review #1 (continued)**
>
> 3)  If the Graph2Seq is particularly important for Natural Question Generation, the author should clarify it more. If the Graph2Seq model is generally applicable to replace the Seq2Seq model, the author should experiment with more tasks. The paper seems not well motivated.
>
> In general, Graph2Seq model should perform better than Seq2Seq model when inputs are actually better represented in graph-structured data. Graph2Seq model has already been proven to perform better than Seq2Seq model in some NLP tasks such as AMR-to-text [1], SQL-to-text [2], and machine translation [3]. However, proving Graph2Seq generally outperforming over Seq2Seq model is out of the scope this paper tried to study.
>
> Instead, in this work, we specifically focus on an important NLP task - Question Generation (QG). We show that it is beneficial to treat the input passage sequence as a passage graph in order to additionally utilize the rich text structure information (both syntactically and semantically), beyond the simple word sequence. We think our findings are particularly useful for QG community to shift the main backbone model choice Seq2Seq to the new arising backbone model Graph2Seq. Our experimental results in Table 1 and 3 have demonstrated the advantages of Graph2Seq over Seq2Seq in QG task.
>
> We will add more discussions on the motivation of using graph structured input and the connection between dynamic graph construction and multi-head attention in transformer in the revision.
>
> [1] Linfeng Song, Yue Zhang, Zhiguo Wang, and Daniel Gildea.  A graph-to-sequence model for amr-to-text generation.arXiv preprint arXiv:1805.02473.
> [2] Kun Xu, Lingfei Wu, Zhiguo Wang, Mo Yu, Liwei Chen, and Vadim Sheinin. Sql-to-text generationwith graph-to-sequence model.arXiv preprint arXiv:1809.05255.
> [3] Joost Bastings, Ivan Titov, Wilker Aziz, Diego Marcheggiani, and Khalil Sima’an.  Graph convolu-tional encoders for syntax-aware neural machine translation.arXiv preprint arXiv:1704.04675,2017.

---

### Author Response · Authors · 2019-11-12
**Response to all reviewers - updated the Q-BLEU1 results**

Dear reviewers,

We were notified by the author of the Q-BLEU1 script (https://github.com/PrekshaNema25/Answerability-Metric) that there was a bug in the old script we used. So we recomputed the Q-BLEU1 scores with the new script on all models and updated the corresponding results in previous responses. We copy the new results below.

Squad split 1
Methods \ Metrics | BLEU-4 | METEOR | ROUGE-L | Q-BLEU1
Transformer-based seq2seq | 2.63 | 7.13 | 24.20 | 15.1
MPQG+R (Song et al., 2017) | 14.39 | 18.99 | 42.46 | 52.0
G2S_dyn + BERT + RL | 17.55 | 21.42 | 45.59 | 55.4
G2S_sta + BERT + RL | 17.94 | 21.76 | 46.02 | 55.6

Squad split 2
Methods \ Metrics | BLEU-4 | METEOR | ROUGE-L | Q-BLEU1
Transformer-based seq2seq | 2.52 | 7.25 | 25.77 | 18.0
NQG (Zhou et al., 2017) | 12.23 | 17.91 | 40.29 | 49.8
MPQG+R (Song et al., 2017) | 14.71 | 18.93 | 42.6 | 50.3
G2S_dyn + BERT + RL | 18.06 | 21.53 | 45.91 | 55.0
G2S_sta + BERT + RL | 18.30 | 21.70 | 45.98 | 55.2

---

### Public Comment · ~Shiyue_Zhang1 · 2019-12-22
**Two related works**

Hi authors,
Thanks for this interesting work for using graph2seq for QG!
I'd like to point two missing related works, which possibly were considered as concurrent works.
One is from Microsoft that using pre-trained seq2seq to improve QG, which, to my best knowledge, is currently the best model on SQuAD-QG (https://arxiv.org/abs/1905.03197, published on NeurIPS-19); the other is my EMNLP-2019 paper, which also applied RL for QG (https://arxiv.org/abs/1909.06356).

---

> ### Author Response · Authors · 2019-12-22
> **Response to "Two related works"**
>
> Hi Shiyue,
> Thank you for pointing out the two concurrent related works which we were previously unaware. We are happy to see there are so many interesting works on this topic. We will acknowledge the above two works in the camera-ready version of our paper.

---

### Decision · Program_Chairs · 2019-12-19

**Decision:**

Accept (Poster)

**Comment:**

The reviewers found this paper on improving NLG using a graph-to-sequence architecture interesting and the results impressive. While I would personally have preferred to see further evaluation of this model on another NLG task, I think it would be overstepping in my role as AC to go against the reviewer consensus. The paper is clearly acceptable.